# The Covid-19 pandemic in Sweden: Prolonged and unevenly distributed effects on the volume of pediatric anesthesia and surgery demonstrated by data from the Swedish Perioperative Register

Sixten Melander[1]*, Gunnar Enlund[2], Helene Engstrand Lilja[3], Peter Frykholm[1]

**1** Department of Surgical Sciences, Section of Anaesthesiology and Intensive Care Medicine, Uppsala University, Uppsala, Sweden, **2** Department of Anaesthesia and Intensive Care, Uppsala University Hospital, Uppsala, Sweden, **3** Department of Women's and Children's Health, Karolinska Institutet, Stockholm, Sweden

* sixten.melander@gmail.com

## Abstract

### Background

In 2020, Covid-19 pushed Swedish health care to its limits regarding access to hospital beds and staffing. A previous investigation of the effects of the first wave of the pandemic in the spring of 2020 revealed a substantial reduction in elective pediatric surgery. The aim of the present study was to expand this analysis on a national and regional level during almost three years with Covid-19.

### Methods

For this retrospective cohort study, routine data from all procedures in patients <16 years of age in 2019–2022 were extracted from the Swedish Perioperative Register. Data were analyzed according to level of care, type of surgery, procedure code and emergency or elective surgery.

### Results

During 2020–2022, the number of surgeries registered was 19,944 fewer than expected as compared to pre-pandemic levels, i.e., a reduction of about 12%. Elective surgery showed a total reduction of 17% while emergency surgery was unaffected. The most dramatic decrease was found in county hospitals where elective surgery was reduced by 28% and the largest effect was found in Ear, Nose, and Throat/oral surgery (−34%). Patient age at the time of surgery did not show any notable differences in total, except for grommets insertion in 2021 and adenoidectomy in 2021 and 2022 compared to 2019.

**Data availability statement:** The Swedish Ethical Review Authority restricts sharing of data since they contain potentially sensitive information. The data is managed by the Swedish Perioperative Register at the Uppsala Clinical Research Centre. Contact person for data requests is Beata Pajak UCR | Uppsala Clinical Research Center Uppsala Science Park, Hubben Dag Hammarskjölds väg 38 751 85 UPPSALA SWEDEN beata.pajak@ucr.uu.se.

**Funding:** The author(s) received no specific funding for this work.

**Competing interests:** The authors have declared that no competing interests exist.

## Conclusion

The Covid-19 pandemic affected the number of surgical procedures in children for more than two years. Future studies of the long-term effects of the large number of canceled operations are warranted.

## Introduction

During the Covid-19 pandemic, the health care service had to redirect its focus and adjust to expand the capacity of intensive care units (ICU) [1]. These adjustments affected different facilities and disciplines according to a report from the National Board of Health and Welfare (NBHW) [2]. From both anesthesiological and surgical perspectives, substantial changes have been reported, such as canceled operations, creation of temporary Covid-related wards, and anesthetic personnel relocated to the ICU [3].

According to another report from the NBHW [4,5], there have been four major waves of Covid-19 in Sweden. The first wave was defined to encompass March – September 2020, the second wave October 2020 – January 2021, the third wave February – June 2021 and the fourth wave July 2021 – March 2022 respectively.

Children are less likely to develop severe Covid-19 disease than adults [6]. The pandemic could still directly or indirectly have had significant effects on pediatric health care services. Cancellation of surgery leading to an increased age at surgery is one example, and it could potentially have affected children's health regarding both short and long-term aspects. We have previously reported that pediatric elective procedures were reduced by more than 50% during the first wave of the pandemic in 2020 [7]. There is a paucity of data concerning the continued effects on pediatric hospitalization and surgery. Therefore, we decided to investigate patterns in the reduction of pediatric procedures on national and regional levels during almost three years with Covid-19. We also aimed to explore the possibility of postponed surgery leading to children being operated at an older age. We hypothesized that the number of cases would continue to be reduced in spite of less hospitalization due to Covid-19 during the studied period.

## Methods

This is a retrospective cohort study based on data extracted from the Swedish Perioperative Register (SPOR). SPOR does not take responsibility for the methods, analysis and results, and the views expressed in this study may not necessarily reflect those of SPOR. Informed consent was waived by the Swedish Ethics Review Authority (permission no 2020−01909). Data was accessed on March 1st 2023. The dataset from SPOR used in this study did not contain complete ID information for individual patients. To be precise, we had information regarding the age at the time of surgery expressed in years and months but not days. Patient name or other identification markers were not available to our research group. The report contained all procedures registered by SPOR from January 1st 2019 to December 31st 2022. The

register includes all procedures performed with anesthesia services involved, i.e., general anesthesia but also diagnostic procedures or treatments with sedation.

## Data sources

SPOR was initiated by the Swedish Society for Anaesthesia and Intensive Care, set up in 2011, and started including data in 2011. All data is managed by Uppsala Clinical Research Centre, which runs many of the largest nationwide quality registers in Sweden. The purpose of SPOR is to provide data for national and local quality assurance projects and research. It is not used for billing purposes. Data are uploaded daily through smooth integration with local surgery planning and billing systems. A process for validation of data is ongoing since several years.

The Swedish Intensive Care Register (SIR) has a similar design and was initiated in 2001 by the same professional organization. The purpose of SIR is to promote and develop quality in Swedish intensive care. Both the above registers have Certification Level 1 and publish open access reports continuously on their respective websites.

## Inclusion and exclusion criteria

Only children <16 years of age at the date of surgery were included, and centers that had not started reporting January 1st 2019 or stopped reporting to SPOR after 2019 were excluded, to enable comparisons with pre-pandemic conditions. We decided early in the planning of the study to only include centers that reported data all four years. All hospitals connected to SPOR upload their data automatically to the register monthly. The uploading process has been validated for all data fields used in this study [8]. Data consist of patient ID, type of surgery, diagnoses, time stamps during the perioperative process (from the decision to operate to the time of discharge from the postoperative recovery area) as well as quality measures.

The year 2019 was defined as baseline for analyses of changes in caseload. The absolute number of cases as well as percentage changes relative to the corresponding baseline periods were reported.

## Outcomes

The primary outcome was reductions in total number of procedures during the pandemic years of 2020, 2021 and 2022 compared to corresponding pre-pandemic numbers, i.e., during 2019. Secondary outcomes were differences in caseload reduction regarding 1) level of care (university, district, and county hospital, as indicated in SPOR; university hospitals are tertiary centers with 24–7 services for most pediatric surgical and medical services, district hospitals have no separate pediatric anesthesia or surgery but capacity to treat children 24–7, county hospitals have only core medical and surgical services and the term smaller units is reserved for private clinics typically focused on elective surgery in a single speciality), 2) surgical specialty according to the national surgical procedure code system KVA-97 [9], 3) emergency vs elective surgery, and 4) age at the time of surgery. For the primary outcome, we determined the weekly caseload and compared to the corresponding week in 2019 to study the evolution of the effects of the pandemic over time. To put these changes into context of the fluctuating intensity of the pandemic we retrieved the number of new admissions to all Swedish Intensive Care Units (ICU) from the Swedish ICU Register [10]. Furthermore, a sub-analysis of the changes during the four different waves of the pandemic was performed, comparing them to baseline numbers of 2019 and displaying differences in number of procedures per day including separate data on emergency and elective procedures.

Finally, out of a total of 3,276 different registered procedures we analyzed mean age at the time of surgery for the 20 most common procedures in 2019 for each of the different years of the study period, to provide more detail and probe for potentially delayed scheduling. Regional distribution as well as changes in waiting times during the pandemic were investigated.

## Confounders

Possible confounders are, e.g., demographic changes such as increasing immigration and variations in nativity, the increasing cost of health care both before and during the pandemic, and possible changes in political policy. These were not accounted for.

## Statistical methods

The dataset is a convenience sample based on data available at the time of study planning. Differences are displayed as absolute and percentage reductions. Poisson regression was used to analyze the main categories during the different years adjusted for risk population based on the population <16 years of age in Sweden during each year using data from Statistics Sweden [11]. Student's t-test was used to investigate changes in patient age at surgery; differences displayed with 95% confidence intervals. A p-value of less than 0.05 was considered statistically significant. Data was managed and analyzed using Microsoft Excel (Version 2210 Build 16.0.15726.20188) and R version 4.1.1.

## Results

The number of procedures was 214,964 reported from 82 centers (14 university hospitals, 20 county hospitals, 45 district hospitals and 3 smaller units). Due to nonreporting during any of the four included years we excluded 2 county hospitals (n = 3,750), 5 district hospitals (n = 394) and 2 smaller units (n = 2,368). The final number of analyzed procedures was thus 208,452. A flowchart of the process is displayed in Fig 1.

The total number of procedures performed in 73 reporting hospitals was 57,099, 49,000, 49,091, and 53,262 procedures in 2019, 2020, 2021 and 2022 respectively (displayed in Table 1). The total reduction during the three pandemic years compared to 2019 as our baseline amounts to 19,944 fewer procedures than what would have been expected without the pandemic, a decrease of approximately 12%. The years 2020 and 2021 showed a similar total reduction of 14%, while 2022 showed a less pronounced reduction of 7%. In Fig. 2, weekly cases are displayed as percentages of the corresponding weeks in 2019, with the number of new ICU-admissions during these 7-day periods superimposed, using data from the Swedish ICU register [10]. The most dramatic reduction in caseload was found in the spring of 2020, with continued, albeit less marked, reductions during the years of 2020, 2021 and 2022 (Table 2). Throughout the pandemic, elective surgery was severely affected while emergency surgery was left largely unchanged. Thus, 40,670, 32,452 (- 20%), 32,123

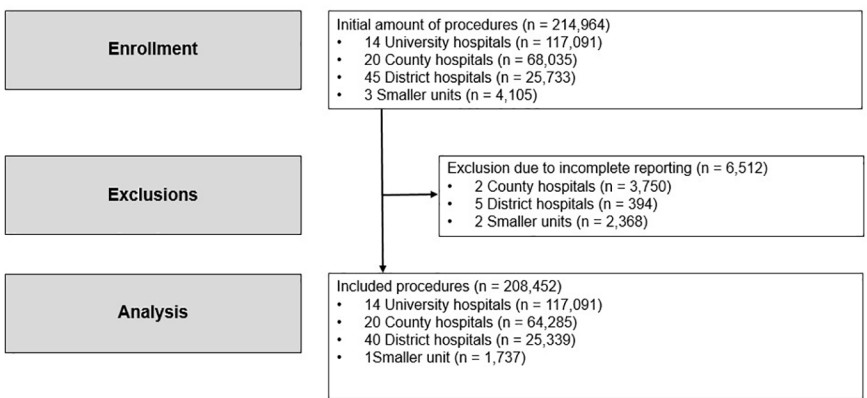

**Fig 1. Cohort flow chart.** The number of procedures enrolled, exclusions due to incomplete reporting during any of the four studied years and total number of procedures included for analysis.

**Table 1. Analyses of number of procedures in hospital categories and main specialties.**

| | | 2019 | 2020 | 2021 | 2022 | Total Difference |
|---|---|---|---|---|---|---|
| | Total | 57,099 | 49,000 (86%, <0.001) | 49,091 (86%, <0.001) | 53,262 (93%. <0.001) | −19,944 (−12%) |
| Total | Emergency | 16,429 | 16,548 (101%, 0.640) | 16,968 (103%, 0.013) | 16,302 (99%, 0.459) | 531 (1%) |
| | Elective | 40,670 | 32,452 (80%, <0.001) | 32,123 (79%, <0.001) | 36,960 (91%, <0.001) | −20,475 (−17%) |
| University | Emergency | 9,376 | 9,446 (101%, 0.716) | 9,789 (104%, 0.008) | 9,249 (99%, 0.337) | 356 (1%) |
| Hospitals | Elective | 20,907 | 18,636 (89%, <0.001) | 18,551 (89%, <0.001) | 21,137 (101%, 0.283) | −4,397 (−7%) |
| County | Emergency | 5,506 | 5,590 (102%, 0.494) | 5,648(103%, 0.280) | 5,524 (95%. 0.886) | 244 (1%) |
| Hospitals | Elective | 13,306 | 9,229 (69%, <0.001) | 8,971 (67%, <0.001) | 10,511 (79%, <0.001) | −11,207 (−28%) |
| District | Emergency | 1,547 | 1,512 (98%, 0.492) | 1,531 (99%, 0.678) | 1,529 (99%, 0.746) | −69 (−1%) |
| Hospitals | Elective | 6,004 | 4,264 (71%, <0.001) | 4,153 (69%, <0.001) | 4,799 (80%, <0.001) | −4,796 (−26%) |
| ENT/Oral | Emergency | 790 | 687 (−87%, 0.007) | 698 (88%, 0.014) | 830 (105%, 0.333) | −155 (−7%) |
| surgery | Elective | 13,829 | 9,230 (67%, <0.001) | 7,884 (57%, <0.001) | 10,256 (74%, <0.001) | −14,117 (−34%) |
| General | Emergency | 3,019 | 3,156 (104%, 0.098) | 3,260 (108%, 0.005) | 2,948 (98%, 0.351) | 307 (3%) |
| Surgery | Elective | 2,447 | 2,116 (86%, <0.001) | 2,025 (83%, <0.001) | 2,208 (90%, <0.001) | −992 (−14%) |
| Orthopedic | Emergency | 6,051 | 5,956 (98%, 0.329) | 5,813 (96%, 0.014) | 5,660 (94%, <0.001) | −724 (4%) |
| Surgery | Elective | 4,616 | 3,907 (85%, <0.001) | 4,204 (91%, <0.001) | 4,614 (100%, 0.967) | −1,123 (−8%) |
| Urological | Emergency | 724 | 777 (107%, 0.188) | 828 (114%, 0.011) | 826 (114%, 0.010) | 259 (12%) |
| Surgery | Elective | 3,241 | 2,627 (81%, <0.001) | 2,634 (81%, <0.001) | 2,723 (84%, <0.001) | −1,739 (−18%) |
| Inhabitants of Sweden | | 1,951,765 | 1,955,716 | 1,961,414 | 1,952,650 | |
| <16 years of age** | | | | | | |

*ENT – Ear, Nose and Throat.

**Information gathered from Statistics Sweden SCB September 8th 2023.

The total number of procedures during 2019–2022 are displayed as emergency and elective categories. Displayed in brackets are firstly their percentage ratios compared to the corresponding numbers in 2019, secondly p-values calculated through rate-ratio comparisons of the number of procedures during the different years, using a Poisson regression, adjusted for the Swedish population at risk during that specific year, using data from Statistics Sweden [11]. Total Difference signifies the sum of difference during the whole pandemic period from start of 2020 to the end of 2022; mean percentage difference per year is displayed in brackets.

(- 21%), and 36,960 (- 11%) elective procedures were registered during 2019, 2020, 2021 and 2022 respectively (Table 1). The corresponding numbers for emergency surgery were 16,429, 16,548, 16,968 and 16,302.

In the rate-ratio analyses, statistically significant changes were found in elective surgery for all specialties and all hospital categories in 2020 and 2021 (last six columns in Table 1). In 2022, significant reductions in elective surgery were found in district and county hospitals but not in university hospitals. Regarding emergency surgery, statistically significant reductions were found for ENT (Ear, nose and throat)/oral surgery in 2020, and in 2021 for university hospitals. University hospitals (n = 14) showed the least marked reduction of elective surgery with a total deficit of 4,397 (7%) over the 3 year period, while county hospitals (n = 18) showed the largest reduction with 11,207 (28%) fewer procedures. District hospitals (n = 40) produced 4,796 (26%) fewer procedures.

Four major groups of surgical specialties were analyzed: ENT/oral surgery, general surgery, orthopedic surgery and urological surgery. As far as elective surgery goes, all specialties showed a reduction ranging from 34% in ENT/oral surgery to 8% in orthopedic surgery. Emergency surgery was reduced by 7% for ENT/oral surgery and by 4% for orthopedic surgery, while general and urological surgery actually increased output by 3% and 12% respectively. In total numbers ENT/oral surgery was most severely affected, a reduction of 14,272 procedures. The corresponding numbers were 1,847 for orthopedic surgery, 1,480 for urological surgery and 685 for general surgery.

Waiting times for adenoidectomy during 2022 are shown for the different regions of Sweden in Fig 3. Waiting times varied between the different regions, with some reporting waiting times of three years or more.

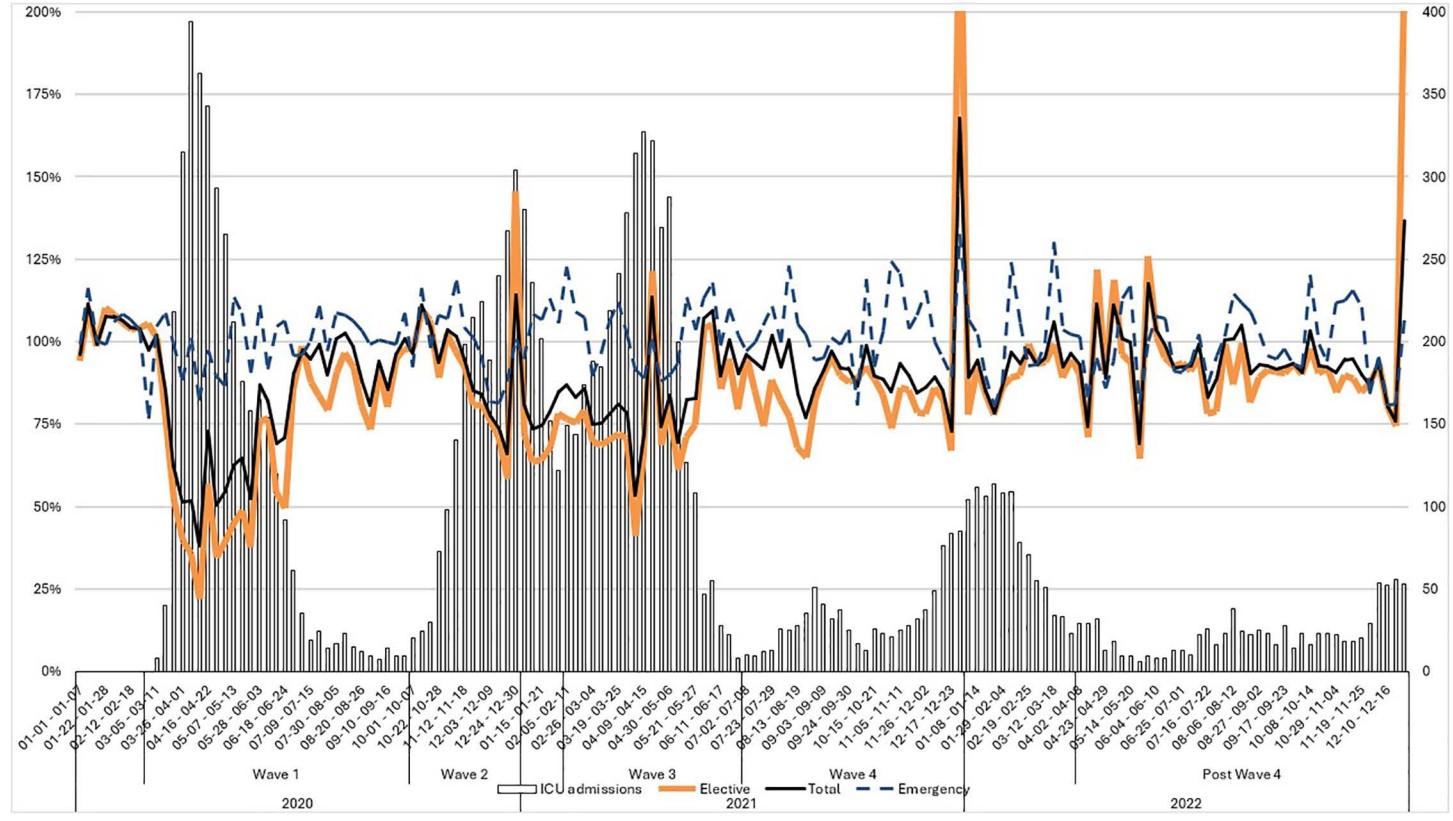

**Fig 2. Cases and ICU admissions throughout the pandemic.** Weekly total (black), elective (blue) and emergency (yellow) cases displayed as percentages of the corresponding weeks in 2019. The number of new ICU-admissions during these 7-day periods are superimposed using data from the Swedish ICU register [10]. New Year's Eve was not included in the 7-day period and February 29th of 2020 was excluded. The official pandemic waves 1 - 4 and year transitions are marked on the x-axis.

There was a small increase in total mean age at surgery during the study period, of approximately two months when comparing 2019 and 2022 (Table 3). Of the 20 most common procedures, we report a statistically significant increase in age from 4.72 (4.57–4.87) years in 2019 to 5.2 (4.94–5.49) in 2021 at the time of grommets insertion, a difference of 6.53 months. Similarly, age at tonsillectomy was 8.27 (8.0–8.57) years in 2019 vs 9.21 (8.76–9.66) in 2021 and 9.46 (9.0–9.91), i.e., mean increases of 11.3 and 14.3 months respectively.

## Discussion

This Swedish register-based study showed that over the course of three full years after the Covid-19 pandemic, the pediatric surgery volume was lower compared to the year before the pandemic. Total caseload remained below historical weekly averages during the entire pandemic, except for a few single weeks. The most dramatic decrease was found during the first wave in the spring of 2020, confirming previously published results [7].

### Emergency vs elective surgery

Elective surgery was severely affected while there was no decline in emergency surgery. There are several plausible explanations for elective surgery cancellations during the pandemic. For example, patients or health care workers having symptoms of Covid-19, personnel relocation or simply postponement of non-urgent surgery to focus on the pandemic.

**Table 2. Number of procedures during the four waves.**

| Wave 1 (March-September 2020) | | | |
|---|---|---|---|
| | Emergency | Elective | Total |
| **2019** | 10,403 | 22,058 | 32,461 |
| **Wave 1** | 10,450 | 14,811 | 25,261 |
| **Difference** | 47 | −7,247 | −7,200 |
| **Difference (%)** | 0.45% | −32.85% | −22.18% |
| **Difference/day** | 0.22 | −33.86 | −33.64 |
| **Wave 2 (October 2020 – January 2021)** | | | |
| | Emergency | Elective | Total |
| **2019** | 4,925 | 15,165 | 20,090 |
| **Wave 2** | 4,924 | 12,325 | 17,249 |
| **Difference** | −1 | −2,840 | −2.841 |
| **Diff (%)** | −0.02% | −18.73% | −14.14% |
| **Difference/day** | −0.01 | −23.09 | −23.10 |
| **Wave 3 (February – June 2021)** | | | |
| | Emergency | Elective | Total |
| **2019** | 6,946 | 17,871 | 24,817 |
| **Wave 3** | 7,159 | 13,924 | 21,083 |
| **Difference** | 213 | −3,947 | −3,734 |
| **Diff (%)** | 3.07% | −22.09% | −15.05% |
| **Difference/day** | 1.41 | −26.14 | −24.73 |
| **Wave 4 (July 2021 – March 2022)** | | | |
| | Emergency | Elective | Total |
| **2019** | 11,840 | 30,177 | 42,017 |
| **Wave 4** | 12,146 | 26,094 | 38,240 |
| **Difference** | 306 163 | −4,083 | −3,777 |
| **Difference (%)** | 2.58% | −13.53% | −8.99% |
| **Difference/day** | 1.12 | −14.90 | −13.78 |

Comparison of the four different waves as defined by NBHW [4,5], displaying Emergency, Elective and Total procedures during the different waves and during corresponding dates in 2019. Also displayed is the total difference and percentage difference between number of procedures compared to corresponding dates in 2019, as well as the difference per day in procedures. Wave 1 lasted for 214 days, wave 2 for 123 days, wave 3 for 151 days and Wave 4 for 274 days.

In addition, recommendations for social distancing may have led to reductions in common pediatric respiratory illnesses [12–15] which could possibly have affected the number of procedures. One Dutch review reported that the Covid-19 pandemic caused significant decrease in pediatric admissions and emergency department (ED) utilization in both the Netherlands and the rest of the world [16]. Furthermore, a German study found that ED consultation during each phase of the pandemic was lower than in pre-pandemic conditions, with larger reductions displayed for less urgent patient consultations and younger patients [17]. This is in line with a British study from the beginning of the pandemic displaying large reductions in ED-visits with the most marked reductions reported in school-age children [18]. A US study found a 42% decline in ED visits during the early pandemic compared to 2019 and found that this reduction was especially pronounced for children and females [19]. Regarding surgical volume during the pandemic a cohort study from a large hospital in Massachusetts found a decline of 44,6% in the early weeks of the pandemic, with the largest differences being displayed in laryngeal, plastic, oral maxillofacial and general surgery while emergent/urgent surgery was less severely affected [20]. This is in line with our results showing more marked reductions in the elective category. There are large spikes in elective

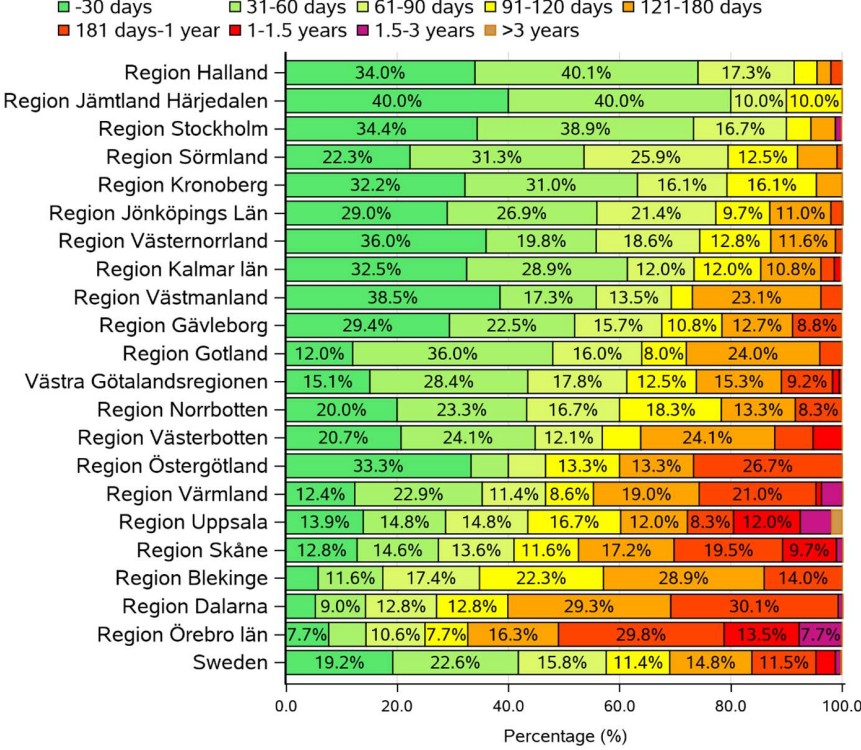

**Fig 3. Distribution of waiting time for adenoidectomy during the pandemic.** Distribution of waiting times for adenoidectomy (EMB30) in the regions of Sweden during 2022. Waiting times were extracted from SPOR by calculating the difference between the time stamp for "Start of surgery" minus the time stamp for "Decision to operate".

surgery in Fig 1 during the last two weeks of 2020, 2021 and 2022. These outliers may be artifacts, explained by the customary low rate of elective surgery during the Christmas season.

### ENT surgery most severely affected

Ear, nose and throat/oral surgery was the group most significantly affected. There could be several explanations for this. Firstly, ENT procedures in children can be postponed for months or even years without serious harm to the children. This is generally the case for, e.g., grommets insertion, tonsillectomy and adenoidectomy, all of which are on the list of the 20 most common procedures (Table 3). However, some children will suffer harm from the consequences of postponing both grommets insertion and tonsil surgery, respectively.

Tonsillectomy has two main indications: sleep-disordered breathing (SBD) and recurrent tonsillitis. In a meta-analysis, SDB was associated with later subsequent development of neurobehavioral deficits (pooled RR 3.24, 95%CI 1.25–8.41) [21]. Furthermore, it has been shown that children with SDB have higher antibiotic usage rates, more hospital visits and more healthcare visits due to upper respiratory infections [22]. The benefits of surgical treatment due to the latter indication are less clear. According to a Cochrane review from 2014, adenoid-/tonsillectomy leads to a reduction in the number of episodes of sore throat and days with sore throat in children in the first year after surgery compared to (initial) non-surgical treatment, although the size of effect of surgery was described as "modest" [23]. Secondly, aerosol-producing otolaryngologic procedures were identified as high-risk early in the pandemic due to the high viral load in the oropharyngeal cavity and adjacent structures, and otolaryngologists were among the first

**Table 3. The 20 most common procedures.**

| Year | 2019 | | 2020 | | 2021 | | 2022 | | Total | |
|---|---|---|---|---|---|---|---|---|---|---|
| Operation | n | Mean Age | n | Mean Age | n | Mean Age | n | Mean Age | n | Mean Age |
| Adenoidectomy | 3,079 | 5.24 (5.12-5.35) | 2,001 | 5.23 (5.09-5.37) | 1,731 | 5.15 (4.99-5.30) | 2,505 | 4.83 (4.71-4.95) | 9,316 | 5.11 (5.04-5.17) |
| Diagnostic exam* | 1,798 | 5.14 (4.96-5.32) | 1,722 | 4.94 (4.76-5.12) | 1,828 | 5.13 (4.95-5.30) | 2,201 | 5.38 (5.21-5.55) | 7,549 | 5.16 (5.07-5.25) |
| Gastroscopy | 1,699 | 9.29 (9.09-9.49) | 1,395 | 9.71 (9.49-9.92) | 1,588 | 9.34 (9.14-9.55) | 1,628 | 9.17 (8.96-9.37) | 6,310 | 9.37 (9.26-9.47) |
| Intracapsular destruction of tonsil | 2,334 | 4.76 (4.66-4.86) | 1,471 | 4.57 (4.45-4.70) | 1,042 | 4.69 (4.53-4.86) | 1,173 | 4.58 (4.43-4.73) | 6,020 | 4.67 (4.60-4.73) |
| Laparoscopic appendectomy | 1,306 | 10.9 (10.8-11.1) | 1,557 | 10.9 (10.7-11.0) | 1,588 | 10.9 (10.7-11.0) | 1,501 | 10.8 (10.7-11.0) | 5,952 | 10.9 (10.8-11.0) |
| MRI Brain | 1,224 | 4.83 (4.61-5.04) | 1,268 | 4.61 (4.39-4.82) | 1,386 | 4.29 (4.11-4.48) | 1,367 | 4.52 (4.32-4.71) | 5,245 | 4.55 (4.45-4.65) |
| Lumbar puncture | 1,475 | 6.64 (6.43-6.85) | 1,267 | 6.59 (6.36-6.82) | 1,253 | 6.04 (5.81-6.28) | 1,053 | 5.84 (5.60-6.08) | 5,048 | 6.31 (6.20-6.43) |
| Tooth extraction | 1,548 | 7.19 (7.03-7.35) | 1,107 | 6.97 (6.78-7.16) | 1,083 | 7.15 (6.97-7.33) | 1,309 | 7.23 (7.05-7.40) | 5,047 | 7.14 (7.05-7.23) |
| Forearm/elbow X,** closed reduction | 1,209 | 8.34 (8.14-8.54) | 1,165 | 8.32 (8.12-8.53) | 979 | 8.42 (8.20-8.65) | 1,067 | 8.42 (8.20-8.63) | 4,420 | 8.37 (8.27−8,48) |
| Grommets | 1,645 | 4.72 (4.57-4.87) | 856 | 4.93 (4.70-5.15) | 517 | 5.21 (4.94-5.49) | 893 | 4.68 (4.48-4.88) | 3,911 | 4.82 (4.72-4.92) |
| Laparoscopic Excision inguinal hernia | 1,061 | 3.46 (3.26-3.67) | 947 | 3.61 (3.39-3.83) | 844 | 3.56 (3.33-3.79) | 891 | 3.68 (3.46-3.91) | 3,743 | 3.58 (3.47-3.68) |
| Operation for undescended or ectopic testis. | 915 | 4.81 (4.56-5.06) | 746 | 5.04 (4.76-5.32) | 786 | 5.14 (4.88-5.41) | 766 | 5.11 (4.83-5.39) | 3,213 | 5.02 (4.88-5.15) |
| Osteosynthesis of forearm or elbowX** | 683 | 8.39 (8.13-8.65) | 697 | 8.06 (7.80-8.32) | 664 | 8.62 (8.35-8.89) | 674 | 8.39 (8.12-8.65) | 2,718 | 8.37 (8.27-8.48) |
| Stereotactic intracranial radiation therapy | 685 | 4.95 (4.74-5.14) | 572 | 5.16 (4.99-5.34) | 489 | 5.14 (4.88-5.41) | 791 | 4.68 (4.53-4.84) | 2,537 | 4.95 (4.85-5.05) |
| Removal of fracture fixation material Forearm/elbow X** | 559 | 9.25 (8.96-9.53) | 572 | 9.20 (8.93-9.48) | 549 | 9.38 (9.10-9.67) | 560 | 9.43 (9.15-9.70) | 2,240 | 9.31 (9.17-9.45) |
| Tonsillectomy | 829 | 8.27 (8.00-8.55) | 508 | 8.85 (8.49-9.22) | 363 | 9.21 (8.76-9.66) | 358 | 9.46 (9.00-9.91) | 2,058 | 8.79 (8.61-8.97) |
| Circumcision | 641 | 8.84 (8.53-9.15) | 468 | 8.99 (8.63-9.35) | 460 | 9.04 (8.67-9.40) | 488 | 8.84 (8.47-9.20) | 2,057 | 8.92 (8.74-9.09) |
| Osteosynthesis of wrist or hand X** | 444 | 10.9 (10.6-11.2) | 480 | 10.4 (10.1-10.7) | 515 | 10.4 (10.1-10.7) | 513 | 11.2 (10.9-11.5) | 1,952 | 10.7 (10.6-10.9) |
| Wrist or hand X** closed reduction | 523 | 9.86 (9.56-10.2) | 491 | 9.83 (9.53-10.1) | 465 | 9.85 (9.56-10.2) | 468 | 10.0 (9.72-10.3) | 1,947 | 9.89 (9.74-10.0) |
| Adenotonsillectomy | 703 | 5.05 (4.82-5.28) | 493 | 4.88 (4.60-5.17) | 339 | 4.63 (4.25-5.00) | 351 | 4.42 (4.05-4.79) | 1,886 | 4.81 (4.66-4.96) |
| Total | 57,099 | 6.84 (6.80-6.88) | 49,000 | 6.89 (6.83-6.95) | 49,091 | 7.00 (6.94-7.06) | 53,262 | 7.01 (6.95-7.07) | 208,452 | 6.93(6.91-6.95) |

*Diagnostic procedures under general anesthesia or sedation performed by anesthesiologists.

**X – fracture.

The 20 most common procedures during our study period, displayed yearly and with totals, showing number of operations (n), mean age at time of operation and 95% confidence intervals displayed in brackets.

physician casualties [24]. This coupled with the fact that children are more prone to respiratory infections which can be hard to distinguish from a Covid-19 infection made surgical planning difficult and elective procedures in children with upper respiratory tract infections (even when Covid-19 was not the culprit) were therefore postponed at least 7 weeks during the pandemic [25].Thirdly, a large quantity of the surgeries are procedures on the tonsils, where infection is one indication for surgery and infections of the upper respiratory tract have been less common during the pandemic because of social distancing, as shown by the Swedish National Board of Health and Welfare and the Public Health Agency of Sweden [14,15].

## Potential costs and benefits of postponing surgery

Postponing surgery may be problematic for several reasons, such as advancement of illness or increased burden of disease. Especially relevant in children is a window of opportunity being lost, e.g., for cochlear implants, orchidopexy or hypospadias surgery. Fortunately, we found no convincing evidence of significantly higher age at surgery for the sum of the 20 most common procedures except for grommets and tonsillectomy (Table 3). The risk of postponing grommets in children is chronic and recurrent ear infections with fluid in the middle ear leading to hearing loss that in turn affects the child's speech and language development [26]. In line with the discussion above about the consequences of postponing the common ENT procedures, surgery for undescended testis should ideally be performed before the age of 1 to avoid increased risk of infertility and malignancy, although the evidence for the latter is a contentious issue [27]. While the overall age at operation for, e.g., undescended or ectopic testis did not increase significantly, it is likely that many of these operations were postponed beyond the age of 12 months in some if not all centers.

However, potential positive effects have been suggested by Gelardi et al. who reported an improvement in symptomatology for children waiting for adenoidectomy, hinting at a reduced need for surgery during the pandemic [28]. Similarly, Marom et al. reported a reduced incidence in acute otitis media episodes during the pandemic, with high spontaneous resolution rates in children associated with the reduction in grommets insertion procedures [29].

## The uneven burdens of a backlog

Cancellations during the pandemic add burden to an already existing health care debt. This might be especially true for procedures with lower priority. This is illustrated in Fig 3, showing waiting times of several years for adenoidectomy in some regions. The regional differences are striking: the waiting times of > 1.5 years ranges from 0 to 51%, in a few regions reaching more than 3 years. We could speculate that the pandemic has aggravated regional issues with unequal access to health care. Moreover, university hospitals were less affected than the smaller district and county hospitals. This could be because of centralization of more complex pediatric cases to university hospitals, while district/county hospitals mainly deal with basic ENT surgery. For example, procedures on neonates and infants are mostly handled by university hospitals in Sweden, and university hospitals display a higher mortality than county and district hospitals [30]. These more complex cases could be less prone to cancellation or postponement.

Health care workers all over the world are still dealing with this post-pandemic backlog while at the same time having to recover from the stress of the pandemic and worker fatigue [31]. Burnout was a problem for healthcare workers even before the pandemic [32] and the increased workload and stress of the pandemic may have added new social and work-related factors that increase risk of burnout. These problems were aggravated in developing countries where vaccination programs had not yet reached the majority of the population. However, these challenges may trigger initiatives to increase workflow [33,34] and reduce late cancellation rates by more rigorous pre-operative screening protocols [12]. Jiang and Carvalho showed an increased OR-efficiency during the pandemic, with lower cancellation rates due to "no show" or "family refusal" [12]. They stipulated mandatory pre-operative Covid-tests causing increased nursing communications and education as a possible explanation. Future studies will be needed to understand the long-term consequences

of three years of backlog of more than 10% of the annual number of procedures, possibly adding on to known issues of hospital staffing shortages causing staff relocations, external recruitments, added hours and lost hours due to illnesses, in Sweden during the pandemic [35].

### Strengths and limitations

The strength of the present study is that it provides data from a national perioperative database including almost all major Swedish hospitals with pediatric anesthesia. However, some important limitations are noted. The study is retrospective and it was not possible for us to adjust for all possible confounders such as changes in demographics, the increasing cost of health care, etc. This limits the ability to draw conclusions regarding cause-effect. Another limitation is that while presently (2025) SPOR covers 100% of Swedish government-run hospitals a few private clinics perform a small number of pediatric cases without reporting their data to SPOR. It is likely that the private clinics were at least as severely affected by the pandemic as government hospitals but their relative contribution to the results would be negligible since their estimated total caseload is a few percent of the total number of pediatric procedures in Sweden. Moreover, the single year of 2019 serves as baseline in the analysis of the volume for the three following years of the pandemic. The reason for using only 2019 as the baseline year was that this was the first year in which almost all Swedish Hospitals had joined the register. Ideally, we would have had access to complete baseline data from at least three years but since the register was still lacking data from several major centers, inclusion of data reported earlier than 2019 would have been less reliable. Theoretically, the production volume could have been unusually high during 2019, causing an incorrect analysis of reduced volume, but we have no reason to believe that this is the case.

Furthermore, we identified two small hospitals with a total caseload of < 1000 cases per year that stopped their reporting after 2021 and were therefore excluded. We have no information if they stopped reporting due to the pandemic *per se* or because of increased centralization or re-organization of patient flow during these years. An additional seven small hospitals were excluded since they joined SPOR after 2019 which precluded comparisons with baseline conditions before the pandemic. Finally, the pandemic was not fully contained within the study period, but our data indicates that caseload was returning to base-line numbers at the end of 2022.

### Conclusions

The Covid-19 pandemic affected the number of surgical procedures in children for more than two years. Future studies of the long-term effects of the large number of canceled operations are warranted.

### Supporting information

**S1 Table. Common pediatric surgeries during the years of the pandemic.** Yearly total number and quota compared to 2019 of five selected common pediatric surgeries in Sweden.
(JPG)

**S1 Fig. Evolution of waiting time for adenoidectomy.** Monthly evolution of waiting time for adenoidectomy (EMB30) shown in days to surgery (blue = mean, gray = median). Waiting time is drawn from data in SPOR by calculating the difference between the time stamp for "Start of surgery" minus the time stamp for "Decision to operate".
(TIF)

**S2 Fig. Evolution of waiting time for surgeries on tonsils.** Monthly evolution of waiting time for surgeries on tonsils (EMB) shown in days to surgery (blue = mean, gray = median). Waiting time is drawn from data in SPOR by calculating the difference between the time stamp for "Start of surgery" minus the time stamp for "Decision to operate".
(TIF)

**S3 Fig. Evolution of waiting time for inguinal hernia surgery.** Monthly evolution of waiting time for inguinal hernia surgery (JAB) shown in days to surgery (blue = mean, gray = median). Waiting time is drawn from data in SPOR by calculating the difference between the time stamp for "Start of surgery" minus the time stamp for "Decision to operate". (TIF)

**S4 Fig. Evolution of waiting time for gastrostomy.** Monthly evolution of waiting time for gastrostomy (JDB) shown in days to surgery (blue = mean, gray = median). Waiting time is drawn from data in SPOR by calculating the difference between the time stamp for "Start of surgery" minus the time stamp for "Decision to operate". (TIF)

**S5 Fig. Evolution of waiting time for orchidopexy.** Monthly evolution of waiting time for orchidopexy (KFH) shown in days to surgery (blue = mean, gray = median). Waiting time is drawn from data in SPOR by calculating the difference between the time stamp for "Start of surgery" minus the time stamp for "Decision to operate". (TIF)

## Acknowledgments

The authors would like to thank Fabian Söderdahl and Johan Bring at Statisticon AB for help with the statistical analysis.

## Author contributions

**Conceptualization:** Sixten Melander, Peter Frykholm, Gunnar Enlund, Helene Engstrand Lilja.

**Data curation:** Sixten Melander.

**Formal analysis:** Sixten Melander.

**Investigation:** Sixten Melander.

**Methodology:** Sixten Melander, Peter Frykholm.

**Project administration:** Peter Frykholm.

**Supervision:** Peter Frykholm, Gunnar Enlund, Helene Engstrand Lilja.

**Visualization:** Peter Frykholm.

**Writing – original draft:** Sixten Melander.

**Writing – review & editing:** Sixten Melander, Peter Frykholm, Gunnar Enlund, Helene Engstrand Lilja.

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
