## [Decision Letter · Decision Letter 0]

5 Jul 2024

Dear Dr. Melander,

Thank you for submitting your manuscript to PLOS ONE. After careful consideration, we feel that it has merit but does not fully meet PLOS ONE’s publication criteria as it currently stands. Therefore, we invite you to submit a revised version of the manuscript that addresses the points raised during the review process.

We look forward to receiving your revised manuscript.

Kind regards,

Andreas Vilhelmsson, Ph.D

Academic Editor

PLOS ONE

Journal Requirements:

2. Thank you for submitting your work to PLOS ONE. At this time, we require the following information in order to proceed with your submission: As you are reporting a retrospective study, please ensure that you have discussed whether all data were fully anonymized before you accessed them. Please include this information in method statement

Reviewers' comments:

Reviewer's Responses to Questions

**Comments to the Author**

1. Is the manuscript technically sound, and do the data support the conclusions?

Reviewer #1: Partly

2. Has the statistical analysis been performed appropriately and rigorously?

Reviewer #1: No

3. Have the authors made all data underlying the findings in their manuscript fully available?

Reviewer #1: Yes

4. Is the manuscript presented in an intelligible fashion and written in standard English?

Reviewer #1: Yes

Reviewer #1: Thank you for providing the opportunity to review this paper, entitled "Swedish pediatric surgery during Covid: A national registry study on the effects of the pandemic on pediatric anesthesia and surgery ". Below, I have provided my evaluation and comments.

I request information about the data source (SPOR) in the method section. How is data recorded into the register? Is it done manually? Are SPOR validated? Most important, to what extent does SPOR cover all operations performed in Sweden? Do centers from both the public and private sectors report to SPOR? Tonsillectomy is often performed in private centers, and sometimes surgery is offered outside the patient's region. Could this impact the validity of the results?

The title contains "…anesthesia and surgery." However, the results do not focus on anesthesia. Are all surgical interventions included in the results performed under anesthesia or specifically general anesthesia? This needs to be clarified.

Only centers that had started reporting to SPOR before 2019 were included. Were all these centers reporting to SPOR throughout the entire period from 2019 to 2022? Or have some centers stopped reporting to the register due to increased workload during the pandemic? This needs to be clarified.

I also request information about the variables retrieved from SPOR and the data arrangement. For example, I find it hard to understand the calculation of the evolution of waiting times. This needs to be clarified in the manuscript.

The large volume of data analyzed means that even small differences can be statistically significant (which could be seen in the analysis of mean age at surgery). Therefore, the results need to be interpreted from the perspective of what is clinically relevant. However, under the statistical section, the significance level that was considered significant is missing.

Several data points appear to be wrong in the tables. For example, the data in Table 1, column 2019, row total-elective, appears to be incorrect (4,067?). The data in Table 3, column Tonsil surgery, row 2019, also seems wrong (1.6?). Please check all data presented in the manuscript.

The discussion is weak and needs to be revised. Some results are briefly discussed, and some statements are poorly substantiated. For instance, the text suggests that ENT surgeries, including tonsillectomy, can be postponed for months or even years without serious harm. The two main indications for tonsil surgery are upper airway obstruction, causing abnormal ventilation during sleep, and infection-related problems such as recurrent tonsillitis. Studies and national guidelines argue that these patients should be prioritized due to short- and long-term consequences of postponed surgery. The discussion section should acknowledge the potential harm of postponed surgeries.

I hope my comments can help!

**Do you want your identity to be public for this peer review?** For information about this choice, including consent withdrawal, please see our Privacy Policy

Reviewer #1: No

---

## [Author Response · Author response to Decision Letter 1]

8 Sep 2024

Dear editors,

We thank the reviewer and editors for their feedback on our manuscript. A rebuttal letter has been uploaded, which aims to respond to the requirements and questions from the journal and the reviewer.

Kindly,

Sixten Melander, corresponding author, on behalf of co-authors

Gunnar Enlund, Helene Engstrand Lilja and Peter Frykholm

---

## [Decision Letter · Decision Letter 1]

15 Mar 2025

Dear Dr. Melander,

Thank you for submitting your manuscript to PLOS ONE. After careful consideration, we feel that it has merit but does not fully meet PLOS ONE’s publication criteria as it currently stands. Therefore, we invite you to submit a revised version of the manuscript that addresses the points raised during the review process.

We look forward to receiving your revised manuscript.

Kind regards,

Andreas Vilhelmsson, Ph.D

Academic Editor

PLOS ONE

Reviewers' comments:

Reviewer's Responses to Questions

**Comments to the Author**

Reviewer #1: All comments have been addressed

Reviewer #2: (No Response)

2. Is the manuscript technically sound, and do the data support the conclusions?

Reviewer #1: Yes

Reviewer #2: Partly

3. Has the statistical analysis been performed appropriately and rigorously?

Reviewer #1: Yes

Reviewer #2: N/A

4. Have the authors made all data underlying the findings in their manuscript fully available?

Reviewer #1: Yes

Reviewer #2: No

5. Is the manuscript presented in an intelligible fashion and written in standard English?

Reviewer #1: Yes

Reviewer #2: Yes

Reviewer #1: (No Response)

Reviewer #2: This manuscript addresses an important and timely topic, contributing to our understanding of long-term healthcare implications post-COVID-19 on healthcare systems. It is a commendable effort to analyze data from a national cohort.

However, significant revisions are necessary to enhance the methodological rigor and ensure the claims are well-supported by the presented data. Specifically, the conclusions regarding long-term effects of COVID-19 require a more cautious interpretation, given the descriptive nature of the retrospective cohort study and potential biases in the analysis. Furthermore, a more robust time series analysis and careful control for confounding factors are critical to substantiate the findings.I do not agree with the concluding claim that the provided data is evidence for long-term effects of COVID-19, at least based on the evidence provided so far. We observe data indicating an increasing overall stress burden on healthcare worldwide due to various reason that also include i.e. changing demographics, rising costs etc.. These factors are and probably cannot addressed in a cohort study based on retrospective secondary data.

Introduction

1) Include more references; ensure all claims are supported by citations.

Methods

1) Report the methods and study characteristics more thoroughly.

2) Explain the rationale for selecting 2019 as the baseline year. Please adjust the research question and objectives accordingly. Elaborate on why comparisons were not made to an average of the years before.

3) Provide details of the study setting, including the exact number of study centers and the variables used for secondary analyses (e.g., level of care).

4) Include comprehensive details of the cohort, including eligibility criteria and the sources and methods of participant selection.

5) Clearly define all outcomes, exposures, predictors, potential confounders, and effect modifiers.

6) Expand on data sources and measurement methods, including a discussion of the registry as a secondary data source and its likely purpose for billing.

7) Describe efforts to address potential sources of bias in the registry or clarify if this is a convenience sample.

8) Explain how quantitative variables were handled in the analyses. If applicable, describe groupings and provide a rationale for the chosen groupings (e.g., why focus on the 20 most common procedures in 2019 rather than overall procedures?).

9) Provide the rationale for investigating the mean age at the time of surgery and how it relates to the study objectives.

10) Include more robust time-series analyses, such as interrupted time-series analysis (Especially given the claim in the conclusion).

10.1) Consider the influence of different COVID-19 waves, comparing them to baseline and different timings ((i.e. https://doi.org/10.1177/014107682096244, https://doi.org/10.1186/s12889-023-15375-7, https://doi.org/10.1371/journal.pgph.0000029).

10.2) Consider causal impact analysis (https://doi.org/10.2147/rmhp.s459307).

13) Explain how missing data were handled in the study.

14) Provide the r source code for the analysis.

Results

15) Clarify whether cases were excluded and provide details about the overall initial sample. Report the number of individuals at each stage of the study (e.g., eligible, examined for eligibility, confirmed eligible, included).

16) If possible, include a cohort flow chart detailing exclusions and other relevant stages.

17) Ensure all tables in the PDF are fully included to allow for a complete evaluation of the results.

18) Align the results with the research question (hypothesis) and objectives. For instance, results such as age characteristics do not fit the stated focus. Adjust the methods and introduction if the scope is broader than the stated patterns of reduction.

18.1) In particular state the rationale behind studying the age. It is not stated in the introduction, methods or discussion what it hast to do with patterns in the reduction of pediatric procedures or if it is a cofactor effecting reduction. The effect of age is not statistically addressed.

19) no infferential results are presented, aside from the statement in the methods section.

Discussion

19) Provide a cautious overall interpretation of the results and discuss their generalizability.

20) Address the effect of comparing one pre-COVID year to multiple post-COVID years.

21) Discussion evaluation is challenging due to missing results (e.g., incomplete tables).

22) Clarify the claim that "the strength of the present study is that it provides detailed data from a national perioperative database including almost all major Swedish hospitals with pediatric anesthesia." Provide evidence to support this claim and elaborate on what "almost all major" entails, as well as its implications for representativeness.

23) Discuss the study's limitations in more detail, particularly regarding the chosen methods. Consider potential sources of bias or imprecision, addressing both their direction and magnitude.

24) Please relate to other studys that study the impact of covid on admission and surgeries.

Overall

24) Reduce the number of figures in the appendix to only those that are directly relevant, I was overwhelmed

25) I cannot fully evaluate the manuscript, as not all data are included (tables do not fit the manuscript).

**Do you want your identity to be public for this peer review?** For information about this choice, including consent withdrawal, please see our Privacy Policy

Reviewer #1: No

Reviewer #2: **Yes: ** Jonas Bienzeisler

---

## [Author Response · Author response to Decision Letter 2]

22 Apr 2025

Response to Reviewers

Dear editors and reviewers,

We thank the reviewers for their feedback on our manuscript. This is a rebuttal letter

that aims to respond to the questions from the reviewers. We respond to the

questions raised one-by-one below.

Kindly,

Sixten Melander, corresponding author, on behalf of co-authors

Gunnar Enlund, Helene Engstrand Lilja and Peter Frykholm

Reviewer #2: This manuscript addresses an important and timely topic, contributing

to our understanding of long-term healthcare implications post-COVID-19 on

healthcare systems. It is a commendable effort to analyze data from a national

cohort.

However, significant revisions are necessary to enhance the methodological rigor and

ensure the claims are well-supported by the presented data. Specifically, the

conclusions regarding long-term effects of COVID-19 require a more cautious

interpretation, given the descriptive nature of the retrospective cohort study and

potential biases in the analysis. Furthermore, a more robust time series analysis and

careful control for confounding factors are critical to substantiate the findings. I do not

agree with the concluding claim that the provided data is evidence for long-term

effects of COVID-19, at least based on the evidence provided so far. We observe

data indicating an increasing overall stress burden on healthcare worldwide due to

various reason that also include i.e. changing demographics, rising costs etc.. These

factors are and probably cannot addressed in a cohort study based on retrospective

secondary data.

Thank you for this comprehensive review with constructive comments and helpful

suggestions. We have revised the conclusion and have added a discussion in the

limitations on the problems of cause and effect and confounding. Obviously, we

cannot provide the complete picture in this retrospective study. We have also added

a rudimentary time series analysis by the addition of data from the official four waves

of the pandemic.

Introduction

1) Include more references; ensure all claims are supported by citations.

We have provided new references, marked in red.

Methods

1) Report the methods and study characteristics more thoroughly.

We have expanded the methods section as suggested.

2) Explain the rationale for selecting 2019 as the baseline year. Please adjust the

research question and objectives accordingly. Elaborate on why comparisons were

not made to an average of the years before.

The reason for using only 2019 as the baseline year was that this was the first year in

which almost all Swedish Hospitals had joined the Registry. Ideally, we would have

had access to complete baseline data from at least three years but since the Register

was still lacking data from several major centers, inclusion of data reported earlier

than 2019 would have been less reliable.

3) Provide details of the study setting, including the exact number of study centers

and the variables used for secondary analyses (e.g., level of care).

We have provided the number of hospitals in each category contributing with data to

SPOR.

4) Include comprehensive details of the cohort, including eligibility criteria and the

sources and methods of participant selection.

SPOR is national registry with the aim of including all Swedish hospitals with

anesthesia and surgery services. All hospitals connected to SPOR were therefore

eligible for inclusion in the present study. We received a complete dataset from

SPOR for the studied period and excluded only those 9 hospitals that had not

reported data from the complete study period. This was mentioned in the limitations.

We decided from the start to exclude children <16 years old, and they were not in our

initial data. A cohort flow chart has been added for clarification.

5) Clearly define all outcomes, exposures, predictors, potential confounders, and

effect modifiers.

We have tried to more clearly define outcomes and confounders in the methods

section.

6) Expand on data sources and measurement methods, including a discussion of the

registry as a secondary data source and its likely purpose for billing.

We have added some text to further describe SPOR.

7) Describe efforts to address potential sources of bias in the registry or clarify if this

is a convenience sample.

We have added the information of convenience sample in statistical methods.

8) Explain how quantitative variables were handled in the analyses. If applicable,

describe groupings and provide a rationale for the chosen groupings (e.g., why focus

on the 20 most common procedures in 2019 rather than overall procedures?).

Quantitative variables have been clearly described and the rationale for groupings

has been expanded in the methods. We did focus on reduction in overall procedures

as the primary outcome. The choice of more detailed analysis of the 20 most

common procedures was arbitrary, but was included to attempt to provide more detail

for common procedures. It would not have been feasible to report data at this level

for each and every type of procedure, as the total number of different procedure

codes was 3,276, most of them with low total numbers of registered entries.

9) Provide the rationale for investigating the mean age at the time of surgery and how

it relates to the study objectives.

The purpose of the study was to investigate patterns and possible effects of the

reduced capacity during the pandemic. Mean age at surgery for a specific procedure

could be a marker for postponing specific elective operations. Since data on when a

procedure is planned and performed were included in the extracted dataset, we also

decided to look at waiting times.

10) Include more robust time-series analyses, such as interrupted time-series

analysis (Especially given the claim in the conclusion).

Regarding time series analysis, Figure 1 describes a time series analysis with weekly

changes in case-load against the backdrop of ICU admissions as a measure of the

intensity of the pandemic. We have added data on changes during the pandemic

waves as a rudimentary intervention time series analysis.

10.1) Consider the influence of different COVID-19 waves, comparing them to

baseline and different timings ((i.e. https://doi.org/10.1177/014107682096244,

https://doi.org/10.1186/s12889-023-15375-7,

https://doi.org/10.1371/journal.pgph.0000029).

Thank you for this suggestion. We have added information on changes during the

waves in Fig 2 and Table 2.

10.2) Consider causal impact analysis (https://doi.org/10.2147/rmhp.s459307).

It was not in the scope of the present study include a causal impact analysis since it

would have required additional data.

13) Explain how missing data were handled in the study.

The SPOR has an excellent automated data collection system with extensive

validation that provides complete datasets for the variables we analysed. SPOR

provides several other interesting variables such as adverse events (including grade)

that are less well validated and therefore prone to varying amounts of missing data.

This is why we focused on core data such as date, age, and type of procedure.

14) Provide the r source code for the analysis.

This is the source code:

library(openxlsx)

infile1 <- "K:/Academy/UU/Frykholm/Indata/Tabell 1 HT23_utan_losen.xlsx"

adb <- read.xlsx(infile1)

orig <- adb

colnames(adb)[3:6] <- paste0("X", colnames(adb)[3:6])

colnames(adb)[c(9,11,13)] <- c("p1", "p2", "p3")

adb$X2020 <- as.numeric(gsub(" |\\(.*", "", adb$X2020))

adb$X2021 <- as.numeric(gsub(" |\\(.*", "", adb$X2021))

adb$X2022 <- as.numeric(gsub(" |\\(.*", "", adb$X2022))

adb$X2022 <- as.numeric(gsub(" |\\(.*", "", adb$X2022))

colnames(adb)[12] <- "19_22"

adb[,c("20_21", "p4", "20_22", "p5", "21_22", "p6")] <- NA

for(i in 1:17){

adb[i,"19_20"] <- poisson.test(c(adb$X2019[i],

adb$X2020[i]),c(adb$X2019[18],adb$X2020[18]))$estimate

adb[i,"p1"] <- poisson.test(c(adb$X2019[i],

adb$X2020[i]),c(adb$X2019[18],adb$X2020[18]))$p.value

adb[i,"19_21"] <- poisson.test(c(adb$X2019[i],

adb$X2021[i]),c(adb$X2019[18],adb$X2021[18]))$estimate

adb[i,"p2"] <- poisson.test(c(adb$X2019[i],

adb$X2021[i]),c(adb$X2019[18],adb$X2021[18]))$p.value

adb[i,"19_22"] <- poisson.test(c(adb$X2019[i],

adb$X2022[i]),c(adb$X2019[18],adb$X2022[18]))$estimate

adb[i,"p3"] <- poisson.test(c(adb$X2019[i],

adb$X2022[i]),c(adb$X2019[18],adb$X2022[18]))$p.value

adb[i,"20_21"] <- poisson.test(c(adb$X2020[i],

adb$X2021[i]),c(adb$X2020[18],adb$X2021[18]))$estimate

adb[i,"p4"] <- poisson.test(c(adb$X2020[i],

adb$X2021[i]),c(adb$X2020[18],adb$X2021[18]))$p.value

adb[i,"20_22"] <- poisson.test(c(adb$X2020[i],

adb$X2022[i]),c(adb$X2020[18],adb$X2022[18]))$estimate

adb[i,"p5"] <- poisson.test(c(+adb$X2020[i],

adb$X2022[i]),c(adb$X2020[18],adb$X2022[18]))$p.value

adb[i,"21_22"] <- poisson.test(c(adb$X2021[i],

adb$X2022[i]),c(adb$X2021[18],adb$X2022[18]))$estimate

adb[i,"p6"] <- poisson.test(c(adb$X2021[i],

adb$X2022[i]),c(adb$X2021[18],adb$X2022[18]))$p.value

}

write.xlsx(adb, "K:/Academy/UU/Frykholm/Poisson_2023-11-09.xlsx")

Results

15) Clarify whether cases were excluded and provide details about the overall initial

sample. Report the number of individuals at each stage of the study (e.g., eligible,

examined for eligibility, confirmed eligible, included).

This information has been added.

16) If possible, include a cohort flow chart detailing exclusions and other relevant

stages.

Thank you for this suggestion. A flow-chart has been added.

17) Ensure all tables in the PDF are fully included to allow for a complete evaluation

of the results.

The tables have been reformatted

18) Align the results with the research question (hypothesis) and objectives. For

instance, results such as age characteristics do not fit the stated focus. Adjust the

methods and introduction if the scope is broader than the stated patterns of

reduction.

We have expanded our methods section, please see answer below for clarification.

18.1) In particular state the rationale behind studying the age. It is not stated in the

introduction, methods or discussion what it hast to do with patterns in the reduction of

pediatric procedures or if it is a cofactor effecting reduction. The effect of age is not

statistically addressed.

We hypothesized that massive postponement of minor surgery could lead to children

having their surgery at an older age. This issue is explored in table 3. We think

confidence intervals is an appropriate way to report this kind of data rather than a

long list of p-values.

19) no infferential results are presented, aside from the statement in the methods

section.

This is not a classic epidemiological study. We had access to data on procedures and

the age and sex of the patients but no data on confounders such as socioeconomic

factors. We did however use poisson regression to compare the pandemic years with

the preceding year regarding the number of performed procedures in relation to the

population during the respective years.

Discussion

19) Provide a cautious overall interpretation of the results and discuss their

generalizability.

We have revised the interpretation and conclusion.

20) Address the effect of comparing one pre-COVID year to multiple post-COVID

years.

Added this discussion to the limitations.

21) Discussion evaluation is challenging due to missing results (e.g., incomplete

tables).

We apologize for this inconvenience. Apparently the tables were truncated at page

margins. This has been addressed in the current version.

22) Clarify the claim that "the strength of the present study is that it provides detailed

data from a national perioperative database including almost all major Swedish

hospitals with pediatric anesthesia." Provide evidence to support this claim and

elaborate on what "almost all major" entails, as well as its implications for

representativeness.

Sweden has a tax-funded healthcare system and the vast majority of pediatric

surgery is performed in government-run hospitals which are 100% covered in SPOR.

A few small private clinics perform office-based procedures in children without

reporting their data to SPOR. We estimate that less than 2000 procedures with GA

are performed annually in Swedish private clinics, which would amount to less than

1% missing data for this reason. We have added info to the limitations section.

23) Discuss the study's limitations in more detail, particularly regarding the chosen

methods. Consider potential sources of bias or imprecision, addressing both their

direction and magnitude.

The limitations section was expanded.

24) Please relate to other studys that study the impact of covid on admission and

surgeries.

We have expanded our discussion on the topic of admissions and volume of surgery

and added four more references.

Overall

24) Reduce the number of figures in the appendix to only those that are directly

relevant, I was overwhelmed

To overwhelm was not our intention. We included these figures to provide detailed

regional data but detailed comparative analysis was not within the scope of the

present paper.

25) I cannot fully evaluate the manuscript, as not all data are included (tables do not

fit the manuscript).

Tables have been reformatted.

---

## [Decision Letter · Decision Letter 2]

31 Jul 2025

Dear Dr. Melander,

Thank you for submitting your manuscript to PLOS ONE. After careful consideration, we feel that it has merit but does not fully meet PLOS ONE’s publication criteria as it currently stands. Therefore, we invite you to submit a revised version of the manuscript that addresses the points raised during the review process.

We look forward to receiving your revised manuscript.

Kind regards,

Andreas Vilhelmsson, Ph.D

Academic Editor

PLOS ONE

Journal Requirements:

Reviewers' comments:

Reviewer's Responses to Questions

**Comments to the Author**

Reviewer #3: All comments have been addressed

Reviewer #4: (No Response)

2. Is the manuscript technically sound, and do the data support the conclusions?

Reviewer #3: Yes

Reviewer #4: Partly

3. Has the statistical analysis been performed appropriately and rigorously?

Reviewer #3: Yes

Reviewer #4: No

4. Have the authors made all data underlying the findings in their manuscript fully available?

Reviewer #3: Yes

Reviewer #4: Yes

5. Is the manuscript presented in an intelligible fashion and written in standard English?

Reviewer #3: Yes

Reviewer #4: No

Reviewer #3: The paper has addressed the issues previously raised and is now of sufficient quality for publication.

Reviewer #4: This manuscript investigates the changes in surgical volume in Sweden before and after the COVID-19 pandemic. The topic is of considerable interest, and the reported decrease in elective procedures aligns with many previous studies.

However, the submitted manuscript is very difficult to read and review. The authors use numerous unconventional abbreviations, particularly in the tables, and there is no clear distinction between the table legends and the main text, which severely hinders readability and interpretation.

In addition, the right margin of the manuscript is filled with unnecessary grey space created by Microsoft Word's comment function, which compresses the body text and makes it even more difficult to read. The manuscript also contains many typographical errors and nonstandard terminology, and it clearly should have undergone professional editing by a native English speaker before submission.

I strongly recommend that the authors revise the manuscript with greater consideration for reviewers, who are volunteering their time and effort. The manuscript should be written in a way that facilitates the review process.

Below are some specific questions and concerns regarding the manuscript:

Please specify the exact dates that define the "four different waves" of the pandemic. How were these definitions established?

In Table 1, please provide an explanation for the labels "19_20," "19_21," and "19_22." The legend states: "rate-ratio comparisons of the number of procedures during the different years, using a Poisson regression, adjusted for the Swedish population at risk during that specific year, using data from Statistics Sweden." However, it seems that these are simply the inverse of the proportion values. For example, the first entry (1.17 for 19_20) appears to be calculated by dividing 57099 by 49000, which corresponds to the inverse of 86% shown next to 49000. What is the actual basis for the reported p-values?

Does "EMB30" stand for "Distribution of waiting times for adenoidectomy"? Please clarify.

In the phrase "Forearm or elbow X. Closed reduction," what does "X" mean? This expression appears multiple times but is not standard terminology.

Does “extraction implant” mean “implant removal”? Please use established surgical terminology.

Does “Lap. excision” refer to laparoscopic excision? Please spell it out.

“Retentio testis” is a diagnosis, not a surgical procedure.

What is meant by "Diagnostic exam"? Please specify what kind of diagnostic procedure is being referred to.

Table 3 contains misaligned indentations and spacing, making it difficult to read. Please reconstruct the table for clarity.

"ENT" in "ENT/oral surgery" is not an internationally standardized abbreviation. Please provide the full term and its definition in both the abstract and the main text.

The Discussion section contains too many paragraphs. Some paragraphs consist of only one sentence, and others are divided without logical flow. This makes the text difficult to follow and does not adhere to standard academic writing conventions. Please restructure the Discussion for better coherence and readability.

The authors state that "The strength of the present study is that it provides detailed data from a national perioperative database including almost all major Swedish hospitals with pediatric anesthesia." However, this claim is not substantiated by the content of the manuscript. The level of detail in the data is not sufficient to be considered a particular strength.

In the Discussion, the phrase "nursing staff shortages in the wake of the pandemic" is used. Does this imply that there was a substantial reduction in the number of nurses in Sweden due to the pandemic? Please elaborate and provide references if possible.

**Do you want your identity to be public for this peer review?** For information about this choice, including consent withdrawal, please see our Privacy Policy

Reviewer #3: No

Reviewer #4: No

---

## [Author Response · Author response to Decision Letter 3]

13 Sep 2025

Response to Reviewers

Dear editor and reviewers,

We thank the reviewers for their feedback on our manuscript. Please find below in italics our replies the questions and comments from the reviewers.

Kindly,

Sixten Melander, corresponding author, on behalf of co-authors

Gunnar Enlund, Helene Engstrand Lilja and Peter Frykholm

1. This manuscript investigates the changes in surgical volume in Sweden before and after the COVID-19 pandemic. The topic is of considerable interest, and the reported decrease in elective procedures aligns with many previous studies.

However, the submitted manuscript is very difficult to read and review. The authors use numerous unconventional abbreviations, particularly in the tables, and there is no clear distinction between the table legends and the main text, which severely hinders readability and interpretation.

In addition, the right margin of the manuscript is filled with unnecessary grey space created by Microsoft Word's comment function, which compresses the body text and makes it even more difficult to read. The manuscript also contains many typographical errors and nonstandard terminology, and it clearly should have undergone professional editing by a native English speaker before submission.

We apologise for the unfortunate formatting errors and unusual abbreviations. We have revised accordingly.

Specifically, we have limited abbreviations, decreased font size in table legend (which follows PLOS1 formatting guidelines) and updated table titles.

2. Please specify the exact dates that define the "four different waves" of the pandemic. How were these definitions established?

As stated in the 2nd paragraph of the introduction, the definition of the waves was taken from a report from the Swedish NBHW: “According to another report from the NBHW [4], there have been four major waves of Covid-19 in Sweden. The first wave was defined to encompass March - September 2020, the second wave October 2020 - January 2021, the third wave February - June 2021 and the fourth wave July 2021 - April 2022 respectively.”

Upon further review of this reference, a slight error has been noted. The fourth wave was defined as July 2021 – March 2022, meaning we incorrectly included April in our analysis of the fourth wave. Our text as well as the numbers in table 2 have been corrected. The differences are marginal and does not change our conclusions. We have also corrected the timeline on the x-axis in Fig 2.

The citation in the text is as follows, although it is written in Swedish:

4. Uppdrag att stödja regionernas hantering av uppdämda vårdbehov samt följa och analysera väntetider i hälso- och sjukvården. Regeringsbeslut 2020-06-25. Dnr S2020/05634/FS [cited 2025 Dec 5] Available from: https://www.regeringen.se/contentassets/67003d0bc0c54da68d5446c5de6dfb59/uppdrag-att-stodja-regionernas-hantering-av-uppdamda-vardbehov-web.pdf

For clarification we added another citation regarding the definitions of the waves, from the Swedish NBHW:

5. Regeringsuppdrag att stödja regionernas hantering av uppdämda vårdbehov samt följa och analysera väntetider i hälso- och sjukvården – Slutrapport mars 2022. [Internet]. Socialstyrelsen. [cited 2025 Sept 1]. Available from: https://www.socialstyrelsen.se/contentassets/af10f61ee18245dbb1543cd1773002e3/ 2022-3-7798.pd

3. In Table 1, please provide an explanation for the labels "19_20," "19_21," and "19_22." The legend states: "rate-ratio comparisons of the number of procedures during the different years, using a Poisson regression, adjusted for the Swedish population at risk during that specific year, using data from Statistics Sweden." However, it seems that these are simply the inverse of the proportion values. For example, the first entry (1.17 for 19_20) appears to be calculated by dividing 57099 by 49000, which corresponds to the inverse of 86% shown next to 49000. What is the actual basis for the reported p-values?

We have updated table 1 and its legend, removing the inverse calculations and putting the p-values in the same cells as the number of procedures and the percentage ratios for clarification purposes. The labels "19_20," "19_21," and "19_22." have been removed. The p-values were calculated as stated above and in the statistical methods.

4. Does "EMB30" stand for "Distribution of waiting times for adenoidectomy"? Please clarify.

In the phrase "Forearm or elbow X. Closed reduction," what does "X" mean? This expression appears multiple times but is not standard terminology.

Does “extraction implant” mean “implant removal”? Please use established surgical terminology.

Does “Lap. excision” refer to laparoscopic excision? Please spell it out.

“Retentio testis” is a diagnosis, not a surgical procedure.

What is meant by "Diagnostic exam"? Please specify what kind of diagnostic procedure is being referred to.

Table 3 contains misaligned indentations and spacing, making it difficult to read. Please reconstruct the table for clarity.

"ENT" in "ENT/oral surgery" is not an internationally standardized abbreviation. Please provide the full term and its definition in both the abstract and the main text.

We have clarified these questions in our manuscript and corrected formatting errors within table 3. In short, EMB30 is the swedish surgical code for adenoidectomy and has been removed, “X” is short for “fracture”, “extraction” has been changed to “removal”, “Lap.” Is short for “laparoscopic”, retentio testis has been changed to “Operation for undescended or ectopic testis.”, “Diagnostic exam” is explained in the table legend as “Diagnostic procedures under general anesthesia or sedation performed by anesthesiologists” and “ENT” is now clarified as “Ear, Nose and Throat”.

5. The Discussion section contains too many paragraphs. Some paragraphs consist of only one sentence, and others are divided without logical flow. This makes the text difficult to follow and does not adhere to standard academic writing conventions. Please restructure the Discussion for better coherence and readability.

The Discussion has been updated to improve coherence and flow.

8. The authors state that "The strength of the present study is that it provides detailed data from a national perioperative database including almost all major Swedish hospitals with pediatric anesthesia." However, this claim is not substantiated by the content of the manuscript. The level of detail in the data is not sufficient to be considered a particular strength.

Thank you for noting this, we agree and have removed the word “detailed”.

9. In the Discussion, the phrase "nursing staff shortages in the wake of the pandemic" is used. Does this imply that there was a substantial reduction in the number of nurses in Sweden due to the pandemic? Please elaborate and provide references if possible.

We have elaborated briefly on this topic, adding a reference:

35. Rosenbäck R, Lantz B, Rosén P. Hospital Staffing during the COVID-19 Pandemic in Sweden. Healthcare (Basel). 2022 Oct 21;10(10):2116. doi: 10.3390/healthcare10102116. PMID: 36292563; PMCID: PMC9602433.

---

## [Editor Report · Decision Letter 3]

12 Oct 2025

The Covid-19 pandemic in Sweden: prolonged and unevenly distributed effects on the volume of pediatric anesthesia and surgery demonstrated by data from the Swedish Perioperative Register

PONE-D-24-03764R3

Dear Dr. Melander,

We’re pleased to inform you that your manuscript has been judged scientifically suitable for publication and will be formally accepted for publication once it meets all outstanding technical requirements.

Kind regards,

Andreas Vilhelmsson, Ph.D

Academic Editor

PLOS ONE
---

## [Editor Report · Acceptance letter]

PONE-D-24-03764R3

PLOS ONE

Dear Dr. Melander,

I'm pleased to inform you that your manuscript has been deemed suitable for publication in PLOS ONE. Congratulations! Your manuscript is now being handed over to our production team.

Kind regards,

on behalf of

Dr Andreas Vilhelmsson

Academic Editor

PLOS ONE